# Research on RTD Fluxgate Induction Signal Denoising Method Based on Particle Swarm Optimization Wavelet Neural Network

**DOI:** 10.3390/s25020482

**Published:** 2025-01-16

**Authors:** Xu Hu, Na Pang, Haibo Guo, Rui Wang, Fei Li, Guo Li

**Affiliations:** College of Computer Science and Technology, Beihua University, No. 3999 East Binjiang Road, Jilin 132013, China; 15549665225@163.com (X.H.); guohaibo218@163.com (H.G.); wangrui@beihua.edu.cn (R.W.); li_fei2024w@163.com (F.L.); 13258855636@163.com (G.L.)

**Keywords:** RTD fluxgate sensor, particle swarm optimization, wavelet neural network, noise suppression

## Abstract

Aeromagnetic surveying technology detects minute variations in Earth’s magnetic field and is essential for geological studies, environmental monitoring, and resource exploration. Compared to conventional methods, residence time difference (RTD) fluxgate sensors deployed on unmanned aerial vehicles (UAVs) offer increased flexibility in complex terrains. However, measurement accuracy and reliability are adversely affected by environmental and sensor noise, including Barkhausen noise. Therefore, we proposed a novel denoising method that integrates Particle Swarm Optimization (PSO) with Wavelet Neural Networks, enhanced by a dynamic compression factor and an adaptive adjustment strategy. This approach leverages PSO to fine-tune the Wavelet Neural Network parameters in real time, significantly improving denoising performance and computational efficiency. Experimental results indicate that, compared to conventional wavelet transform methods, this approach reduces time difference fluctuation by 23.26%, enhances the signal-to-noise ratio (SNR) by 0.46%, and improves sensor precision and stability. This novel approach to processing RTD fluxgate sensor signals not only strengthens noise suppression and measurement accuracy but also holds significant potential for improving UAV-based geological surveying and environmental monitoring in challenging terrains.

## 1. Introduction

Aeromagnetic survey technology, which measures the Earth’s magnetic field using magnetometers mounted on aircraft, has been widely applied in geological exploration and environmental monitoring [1]. However, due to complex terrain, weather conditions, and geomagnetic interference, traditional techniques often struggle to provide accurate data [2]. To address these limitations, unmanned aerial vehicle (UAV)-based aeromagnetic survey systems equipped with residence time difference (RTD) fluxgate sensors have emerged as promising alternatives, particularly suited for complex and remote terrains [3]. RTD fluxgate sensors, known for their small size, lightweight, and high precision, are ideal for deployment on UAVs. However, the Barkhausen noise originating from the magnetic core of the sensitive element, combined with environmental noise, interferes with time differential detection, thus reducing the sensor’s stability [4,5]. Consequently, developing effective denoising methods is essential for enhancing the precision and stability of these sensors.

In response to these challenges, researchers have focused on enhancing the performance and stability of fluxgate sensors. Internationally, research efforts have primarily focused on optimizing sensor structures, noise suppression technologies, and signal processing algorithms [6,7,8], while domestic research in China has emphasized improving the anti-interference capability and precision of the sensors [9]. Despite advancements in these areas, achieving effective noise suppression, particularly against Barkhausen noise and environmental interference, remains a persistent challenge for RTD fluxgate sensors [10]. Current denoising methods encompass filtering, signal decomposition, and intelligent algorithms [11,12,13]. Filtering techniques can reduce noise to some extent but often result in the loss of useful signals, impacting measurement accuracy. Signal decomposition methods offer advantages in multi-scale signal processing, but their performance is constrained by parameter selection and decomposition accuracy, which are critical in high-precision applications. Intelligent algorithms provide adaptability and learning capability, yet they are prone to becoming trapped in local optima, which limits their effectiveness.

To overcome these limitations, this paper proposes a novel denoising method for RTD fluxgate induction signals, integrating Particle Swarm Optimization (PSO) with Wavelet Neural Network [14,15]. PSO, known for its capability to avoid local optima in optimization tasks, is combined with Wavelet Neural Network advantages in signal decomposition and adaptive processing. By introducing compression factors and dynamic adjustment strategies, this method enables flexible tuning of the parameters and structure of the Wavelet Neural Network, allowing it to better adapt to the complex characteristics of induction signals. Consequently, the proposed approach not only enhances noise reduction but also significantly improves the measurement accuracy and stability of RTD fluxgate sensors. With potential for broad applications, this method represents a new solution for UAV-based aeromagnetic surveys, particularly in geologically challenging and remote areas [16].

## 2. Principle of RTD Fluxgate and Noise in Sensitive Units

The sensitive unit of the RTD fluxgate sensor includes an excitation coil, induction coil, magnetic core, and a supporting frame, as shown in Figure 1. The frame, made of either double-layered semiglass sheets or hollow cylindrical plastic, has a soft magnetic material core sandwiched in the middle. Since the magnetic core needs to possess characteristics such as high magnetic permeability, low coercivity, high-saturation magnetic induction, and rapid saturation, these properties help enhance the sensitivity and response speed of the RTD fluxgate sensor, while reducing energy loss and improving the accuracy and stability of the signal. Therefore, cobalt-based amorphous magnetic cores, iron-based nanocrystalline magnetic cores, or iron-based amorphous magnetic cores can be chosen [17,18]. It is known from experiments that both the Fe-based nanocrystalline magnetic core and the Co-based amorphous magnetic core can reach a saturated state within the excitation period, and there are obvious spike pulses in the induced signals. However, when the Fe-based amorphous magnetic core is used, the pulse width of the induced signal occupies the entire excitation period. Even when the dynamic permeability of the magnetic core is at its maximum, there is still no obvious spike pulse in the induced signal. Therefore, it is not suitable to be used as the magnetic core of the sensitive unit of the time difference fluxgate sensor. Compared to iron-based nanocrystalline magnetic cores, when the Co-based amorphous magnetic core is adopted, the induced signal output by the sensitive unit has the steepest rising edge, the narrowest pulse width, and the highest amplitude, which meets the requirements for the selection of magnetic core materials for the sensitive unit of the time difference fluxgate sensor. The excitation coil is wound around the frame’s ends, while the induction coil is placed centrally around the core, creating a structure that maximizes magnetic responsiveness, as illustrated in Figure 1.

In this experiment, we opted for cobalt-based amorphous magnetic core materials featuring low coercivity and high permeability, which allowed us to obtain hysteresis loops that were very close to the ideal state. To further improve the performance of the time difference fluxgate sensor, the selected magnetic core materials also boasted the Schmidt trigger characteristic, meaning that they could respond promptly to magnetic field changes under saturated conditions. Additionally, in order to enhance the accuracy of magnetic field measurement, the induced voltage pulses should not only possess a relatively high pulse amplitude and a narrow pulse width but also require a steeper rising edge. Through optimizing the design of the sensitive unit, we have successfully established a model in which the characteristics of the magnetic core are close to the ideal hysteresis loop, as shown in Figure 2a. In the figure, point $H_{c}$ represents the coercivity of the magnetic core, point *B* indicates the magnetic flux density, and point $B_{s}$ denotes the saturation magnetic flux density. When the applied axial magnetic field $\left|H_{e}\right|$ is less than the threshold value $H_{c}$, the permeability of the core is μ (for an ideal core, the permeability approaches infinity). However, when $\left|H_{e}\right|$ exceeds $H_{c}$, the magnetic permeability of the core will not immediately drop to zero but gradually approaches zero. This is because when the external magnetic field strength exceeds a certain value, although most magnetic domains are aligned along the direction of the external magnetic field, the material itself has unevenness, such as crystal lattice defects, impurities, etc. These microscopic factors make it difficult for a small number of magnetic domains to be fully aligned along the direction of the external magnetic field, so the magnetic permeability will not drop directly to zero but gradually approaches zero as the external magnetic field is further strengthened. By applying a periodic triangular wave current to the excitation coil, which is sufficient to saturate the magnetic core, a magnetic field is generated within the fluxgate core, as illustrated in Figure 2b. When the external magnetic field under measurement is zero, the time intervals during which the core remains in positive and negative saturation are equal (as shown by the solid line in Figure 2c), resulting in a time difference $\Delta T=T^{+}-T^{-}=0$ (solid line in Figure 2d). However, when an external magnetic field is present, its bias effect causes the time intervals in positive and negative saturation states to become unequal (as shown by the dashed line in Figure 2c), leading to a nonzero time difference $\Delta T=T^{+}-T^{-}\neq 0$ (Figure 2d). Therefore, the magnitude of the time difference can be used to measure the external magnetic field [19].

The RTD fluxgate sensor is susceptible to both environmental and inherent noise during practical applications. Environmental noise primarily stems from external electromagnetic sources, such as power lines and radio waves, which induce fluctuations in the sensor’s output signal, increasing measurement errors and compromising the accuracy and reliability of the results. Additionally, these external interferences can cause nonlinear responses in the sensor, leading to deviations in measurement data and making precise data analysis challenging [20].

Inherent noise, on the other hand, encompasses various types such as excitation noise, magnetic noise, and induction noise. Excitation noise, caused by different excitation waveforms, affects the sensor’s response speed and sensitivity, and excessive excitation noise may result in a signal delay or distortion, impairing both the real-time performance and accuracy of measurements. Furthermore, it limits the sensor’s dynamic range, thereby constraining sensitivity and precision. Magnetic noise arises mainly from the sensor’s internal magnetic materials and external magnetic field fluctuations, leading to signal instability and increased measurement errors. Induction noise is generated by changes in both the sensor’s internal electronic components and external electromagnetic fields, degrading signal quality and increasing measurement uncertainty.

To enhance the sensor’s performance and the reliability of measurement outcomes, it is crucial to adopt effective denoising methods that can mitigate these noise impacts. Existing denoising methods, however, often fall short in effectively addressing these challenges, especially in complex environments. Therefore, this study proposes a novel denoising approach that leverages a PSO-enhanced Wavelet Neural Network, aimed at improving signal accuracy and reliability [21].

## 3. The Denoising Model for Induced Signals Based on PSO–Wavelet Neural Network

### 3.1. Wavelet Neural Network

Compared to traditional BP neural networks and other feedforward neural networks, a Wavelet Neural Network demonstrates superior learning capabilities, faster convergence rates, and greater precision. These advantages, along with heightened sensitivity in function approximation and robust fault tolerance, make Wavelet Neural Networks particularly effective in addressing complex signal denoising tasks.

In this study, we leverage the strengths of Wavelet Neural Networks by integrating wavelet transforms’ multi-scale analysis with neural networks’ non-linear processing ability. This hybrid approach allows Wavelet Neural Networks to adaptively capture signal variations across different scales, enabling efficient handling of both high-frequency and low-frequency components in RTD fluxgate sensor signals. As a result, this method enhances the accuracy and stability of denoised signals, proving especially suitable for applications that demand high precision in noise suppression and signal fidelity, such as UAV-based geological surveying and environmental monitoring.

The topological structure of the Wavelet Neural Network is illustrated in Figure 3. In the figure, *X*_1_*, X*_2_, …, *X_m_* represent the input parameters of the Wavelet Neural Network, while *Y*_1_, *Y*_2_, …, *Y_l_* denote the predicted output values. $\omega_{j\;k}$ and $\omega_{i\;j}$ refer to the connection weights between the input layer and the hidden layer and between the hidden layer and the output layer, respectively [22,23].(1)$$h(j)=h_{j}(\frac{\sum_{i=1}^{n}\omega_{ij}x_{i}-b_{j}}{a_{i}})\;j=1,2,\ldots ,l$$

When the input signal sequence is *X_i_* (*i* = 1, 2, …, *k*), the output formula for the hidden layer is as follows:

In Formula (1): $h\left(j\right)$ is the output value of the j node in the hidden layer. $h_{j}$ is the wavelet basis function. $a_{i}$ is the scaling factor of the wavelet basis function $h_{j}$. $b_{j}$ is the translation factor of the wavelet basis function $h_{j}$. The calculation formula of the output layer is as follows:(2)$$y(k)=\sum_{i=1}^{l}\omega_{jk}h(i)\;\;k=1,2,\ldots ,m$$

In Formula (2): h(i) is the output value of the i hidden layer. l is the number of hidden layer nodes. m is the number of output layer nodes.

The Wavelet Neural Network usually uses the gradient correction method to correct the network weights and wavelet basis function parameters. The correction process is as follows:

Calculate the prediction error of Wavelet Neural Network:(3)$$e=\sum_{k=1}^{m}yn\left(k\right)-y\left(k\right)$$

In Formula (3): $yn\left(k\right)$ is the expected output. $y\left(k\right)$ is the predictive output of the Wavelet Neural Network.

Correct the weight of Wavelet Neural Network according to the prediction error:(4)$$\omega_{n,k}^{\left(i+1\right)}=\omega_{n,k}^{i}+\Delta \omega_{n,k}^{\left(i+1\right)}$$

The coefficients of the wavelet basis function are corrected according to the prediction error e:(5)$$a_{k}^{\left(i+1\right)}=a_{k}^{i}+\Delta a_{k}^{\left(i+1\right)}$$(6)$$b_{k}^{\left(i+1\right)}=b_{k}^{i}+\Delta b_{k}^{\left(i+1\right)}$$

In Equations (4)–(6), $\Delta \omega_{n,k}^{\left(i+1\right)}$, $\Delta a_{k}^{\left(i+1\right)}$, and $\Delta b_{k}^{\left(i+1\right)}$ are calculated by the network prediction error. The calculation method is as follows:(7)$$\Delta \omega_{n,k}^{\left(i+1\right)}=-\eta \frac{\partial e}{\partial \omega_{n,k}^{\left(i\right)}}$$(8)$$\Delta a_{k}^{\left(i+1\right)}=-\eta \frac{\partial e}{\partial a_{k}^{\left(i\right)}}$$(9)$$\Delta b_{k}^{\left(i+1\right)}=-\eta \frac{\partial e}{\partial b_{k}^{\left(i\right)}}$$
where *η* is the network learning rate.

### 3.2. PSO Algorithm

In the training process of the traditional Wavelet Neural Network, there is a tendency to fall into local minima, leading to slower convergence rates and difficulty achieving the required error precision. These limitations restrict the effectiveness of the Wavelet Neural Network for denoising induction signals effectively [24]. To improve the convergence speed, training accuracy, and diagnostic precision of the Wavelet Neural Network, a PSO algorithm is introduced to optimize the network structure when constructing the denoising model for RTD fluxgate induction signals [25].

The optimization process starts by obtaining an optimal parameter set for particles using the PSO algorithm. This parameter set is then used to assign initial weights and thresholds between nodes in each layer of the Wavelet Neural Network, thereby optimizing the corresponding translation and scaling factors. By optimizing these elements, the PSO enhances the Wavelet Neural Network capability to isolate and suppress noise, thus improving the accuracy and reliability of RTD fluxgate signal readings in dynamic environments. In PSO, the velocity and position of the particles are generally updated according to the following equations.

The speed update formula is as follows:(10)$$v_{id}^{t+1}=w\cdot v_{id}^{t}+c_{1}\cdot r_{1}\cdot \left(p_{best_{d}}^{t}-x_{id}^{t}\right)+c_{2}\cdot r_{2}\cdot \left(g_{best_{d}}^{t}-x_{id}^{t}\right)$$
where $v_{i}^{t+1}$ is the velocity of the i-th particle at the t+1-th iteration in the d-dimension, w is the inertia weight, $c_{1}$ and $c_{2}$ are learning factors, $r_{1}$ and $r_{2}$ are random numbers between [0, 1], $p_{best_{id}}^{t}$ is the optimal position of the i-th particle in the d-dimension up to the t-th iteration, and $g_{best_{d}}^{t}$ is the optimal position of all particles on the d-dimension up to the t-th iteration.

Location update formula is as follows:(11)$$x_{id}^{t+1}=x_{id}^{t}+v_{id}^{t+1}$$
where $x_{id}^{t+1}$ is the position of the i particle at the t+1 iteration in the d-dimension. Figure 4 is the flow chart of PSO algorithm.

### 3.3. Improvement Strategy for PSO

The Wavelet Neural Network demonstrates strong nonlinear approximation capabilities and adaptability, making it suitable for complex denoising tasks. However, when applied with traditional PSO, these capabilities are not fully leveraged due to the static nature of conventional PSO [26]. By introducing dynamic updating strategies, the parameters and structure of the Wavelet Neural Network can adapt more effectively to the complex characteristics of RTD fluxgate sensor signals, resulting in enhanced denoising performance [27].

The enhancements to the standard PSO algorithm are achieved by adjusting its parameters to balance the algorithm’s global exploration and local exploitation abilities. Specifically, the velocity evolution equation incorporates both inertia weight (ω) and compression factor (χ). These parameters are dynamically adjusted—either linearly or non-linearly—based on the iteration process and particle behavior, balancing global search capabilities with convergence speed. Through these adjustments, the PSO algorithm achieves improved adaptability to the denoising needs of RTD fluxgate sensor signals, thereby optimizing the Wavelet Neural Network performance in terms of noise suppression and measurement accuracy.

The formula for the linear decreasing strategy of dynamically adjusting inertia weight ω is as follows:(12)$$\omega =\omega_{start}-\frac{\left(\omega_{start}-\omega_{end}\right)\cdot t}{T_{max}}$$

In particular, $w_{start}$ and $w_{end}$ are the initial and final values of inertia weight, respectively. t is the current number of iterations. $T_{max}$ is the maximum number of iterations.

Velocity update formulation for PSO is as follows: The velocity and position of the particles are adjusted using a compression factor, allowing for more controlled movement of the particles within the search space. This prevents the particles from moving too far or too fast. The velocity update equation for the PSO algorithm with a compression factor is given as follows:(13)$$v_{id}^{t+1}=\chi \left[v_{id}^{t}+c_{1}\cdot r_{1}\cdot \left(p_{\;best_{id}}^{t}-x_{id}^{t}\right)+c_{2}\cdot r_{2}\cdot \left(g_{\;best_{d}}^{t}-x_{id}^{t}\right)\right]$$

In this context, χ represents the compression factor, the calculation of which is typically based on the sum of the learning factors $c_{1}$ and $c_{2}$, as well as the dimensionality of the particle swarm, *D* (although in certain literature, the dimensionality *D* may not explicitly appear in the formula for the compression factor, it nevertheless influences the algorithm’s performance and parameter selection) [28]. However, a common formula for the compression factor is as follows:(14)$$\chi =\frac{2}{\left|2-\varphi -\sqrt{\varphi^{2}-4\varphi }\right|}$$
where φ is a function of the sum of $c_{1}$ and $c_{2}$ multiplied by them.

### 3.4. Wavelet Neural Network Model Based on Improved PSO

The Wavelet Neural Network model optimized by the improved PSO is designed to enhance model efficiency by adjusting the parameters of the Wavelet Neural Network. The detailed optimization procedure is as follows:

Step 1:

Input the signal data collected by the RTD fluxgate sensor along with the associated noise data. These data are preprocessed through normalization to facilitate the subsequent wavelet transformation and neural network processing stages.

Step 2:

Map the key Wavelet Neural Network parameters, such as network structure and weights, to particle positions in the PSO algorithm. The particle swarm is then initialized, and multiple iterations are conducted to search for the optimal parameter set. The algorithm advances to the next step when either of the following conditions is met: (1) The model’s prediction error falls below a predefined error threshold, or (2) the maximum number of iterations is reached. If neither condition is satisfied, the particle positions and velocities are updated, and the search for the optimal solution continues.

Step 3:

Set the globally optimal values identified by the PSO as the parameters of the neural network. Fitting tests and error evaluations are performed to assess the model’s accuracy. If the test results meet the required accuracy threshold, these globally optimal values are confirmed as the model’s parameters. Otherwise, the algorithm returns to Step 2 to refine the parameter search until the accuracy requirement is fulfilled.

Step 4:

Apply the optimized model to denoise the RTD fluxgate sensor signals, with the aim of enhancing measurement accuracy and stability.

The overall workflow of the Wavelet Neural Network model optimized by the improved PSO algorithm is illustrated in Figure 5.

## 4. Denoising Analysis of Induced Signal

### 4.1. Experimental Design

To validate the effectiveness of the proposed denoising method based on the PSO-enhanced Wavelet Neural Network for RTD fluxgate sensor signals, we designed a series of controlled experiments. We utilized two datasets: one composed of raw magnetometer signals mixed with ambient noise, and the other consisting of denoised signals processed by conventional wavelet transform methods, serving as a baseline for comparison. The denoising performance was evaluated using the signal-to-noise ratio (SNR) and time difference fluctuation metrics, as these directly reflect improvements in signal clarity and measurement stability.

### 4.2. Data Sample Collection

Data were acquired in real time using an RTD fluxgate sensor under an external magnetic field strength of 10,000 nT, with a 200 kHz sampling frequency to capture high-resolution magnetic field variations, crucial for evaluating the proposed denoising method across varying field intensities. This high sampling rate and the precision of the RTD fluxgate sensor provided a detailed dataset for validating the denoising approach.

### 4.3. Experimental Procedures

Data preprocessing included the following: the RTD fluxgate sensor signal data were preprocessed, including normalization, to ensure stability and consistency. This preprocessing step is crucial for minimizing time difference fluctuations and enhancing the reliability of subsequent analysis.

Preliminary denoising using a wavelet transform included the following: A wavelet transform was applied for preliminary denoising. This involved selecting appropriate wavelet basis functions and decomposition levels, allowing for effective high-frequency noise suppression through thresholding. Selection of the wavelet basis considered orthogonality, compact support, symmetry, regularity, and vanishing moments to address the characteristics of Barkhausen noise, which interferes with RTD readings. Table 1 summarizes the chosen wavelet bases and their respective properties.

The Daubechies wavelet (db wavelet) is a kind of orthogonal wavelet with compact support, orthogonality, and good time–frequency localization characteristics. It is a wavelet widely used in signal processing and data compression. By comparing the SNR and first-order difference variance (FDV) of db series wavelets, the appropriate wavelet basis function was selected. Table 2 is as follows:

To achieve better optimization results, wavelet functions with a high SNR and low FDV should be selected. As observed from the table above, db3 and db4 wavelets demonstrate superior performance. In wavelet transforms, db3 and db4 possess three and four vanishing moments, respectively. The number of vanishing moments determines the orthogonality and compact support of the wavelet functions. Additionally, db3 has a filter length of six (i.e., six filter coefficients), while db4 has a filter length of eight. A longer filter length implies higher frequency resolution and improved time–frequency localization capabilities. This suggests that db4 provides higher precision when approximating polynomials, allowing for a more accurate representation of high-frequency components in the signal. Since the fluxgate sensor’s periodic pulse signals contain more high-frequency components or exhibit rapid variations, db4 is a more suitable choice. Its higher number of vanishing moments and longer filter length enable a better representation of signal details.

Regarding the choice of decomposition levels, a 4-level decomposition can effectively capture the main features and structure of the signal without over-decomposing and causing a loss of detail. This allows the primary components of the signal to be retained in the lower frequency bands, while high-frequency noise is decomposed into higher levels, where it can be removed through threshold processing.

Construction of the Wavelet Neural Network model to further denoise the preprocessed signal included the following: The network structure was designed by determining the number of nodes in the input, hidden, and output layers, and the training was performed using the backpropagation algorithm. The relevant initial parameters set during training are shown in Table 3.

Parameterization of the PSO included the following: The improved PSO algorithm was employed to optimize the parameters of the Wavelet Neural Network, including the learning rate, weights, and thresholds. The initial parameters of the PSO were set, and iterative updates were performed to find the optimal solution, aiming to enhance the denoising performance. Specifically, the number of particles in the swarm was set to 30, with the inertia weight parameter set at 0.73, the cognitive parameter at 1.50, and the social parameter also at 1.50. The maximum number of iterations was set to 50.

The results analysis included the following: The signals before and after denoising were analyzed by calculating the SNR and the mean square error (MSE). A comparative analysis of the denoising performance was conducted using charts to illustrate the effectiveness of the method.(15)$$A_{i}=\frac{x_{i}-x_{min}}{x_{max}-x_{min}}\;$$

In the formula above, $A_{i}$ is the normalized input data, $x_{i}$ is the original input data, and $x_{max}$ and $x_{min}$ are the maximum and minimum values in the original input data, respectively.

### 4.4. Results Analysis

The experimental results are shown in Figure 6, Figure 7 and Figure 8. Figure 6 illustrates the denoising performance at the pulse peak of the induction signal with noise, the signal after wavelet transform denoising, and the signal processed by the Wavelet Neural Network optimized by the PSO algorithm.

As depicted in Figure 6, from the perspective of waveform smoothness, the original signal (blue line) exhibits significant fluctuations, indicating the presence of pronounced high-frequency noise. After the wavelet transform, the fluctuations in the signal (purple line) are reduced, demonstrating that wavelet denoising effectively suppresses noise to a certain extent. However, the signal processed by the PSO-enhanced Wavelet Neural Network (red line) is the smoothest, with almost all high-frequency noise eliminated, showing a superior denoising capability.

Regarding noise suppression, the wavelet transform reduces noise by decomposing the signal and removing high-frequency components, though it may encounter signal distortion due to the pseudo-Gibbs phenomenon [29]. In contrast, the PSO-enhanced Wavelet Neural Network method, by optimizing the parameters of the Wavelet Neural Network through the PSO algorithm, can more accurately identify and remove noise while preserving the original signal’s features. Moreover, the signal processed by the PSO-enhanced Wavelet Neural Network retains more detailed characteristics of the original signal while effectively removing noise, resulting in a denoised signal that more closely resembles the true form of the original signal.

The overall denoising performance of the induction signal is shown in Figure 7. The left panel displays the original signal, the middle panel shows the signal after wavelet transform denoising, and the right panel illustrates the signal processed by the PSO-enhanced Wavelet Neural Network:

The upper left figure shows the original signal. It can be observed that the presence of noise significantly affects the clarity and accuracy of the signal. The middle figure presents the signal after denoising using a wavelet transform. As a classical signal processing method, a wavelet transform effectively removes part of the noise; however, upon closer inspection, a certain amount of residual noise remains, particularly at the edges or in the high-frequency components of the signal. The upper right figure illustrates the signal after denoising using the Wavelet Neural Network optimized by the improved PSO algorithm. Comparatively, this method demonstrates a smoother overall denoising performance, with noise being further suppressed. Notably, a better balance is achieved between noise reduction and the preservation of signal details. This indicates that the improved algorithm offers advantages in handling complex noise while retaining critical signal characteristics.

In the experiment, sample variance was chosen as the performance metric to evaluate the effectiveness of the denoising algorithms, as it reflects the volatility of the signal after noise removal. Lower sample variance indicates that the denoising algorithm can more consistently process the signal and reduce noise interference. In contrast, higher sample variance may suggest that a significant amount of noise remains or that too much signal detail has been lost. The data analysis results show that, compared with the original signal, the sample variance of the signal processed by the Wavelet Neural Network optimized by the improved PSO algorithm decreased by 0.24%. In comparison to using only a wavelet transform, the sample variance decreased by 0.22%. The calculated sample variance values are presented in Table 4.

Figure 8 presents the power spectral density (PSD) of the original noisy induction signal, as well as the PSDs after denoising using a wavelet transform and the Wavelet Neural Network optimized by the improved PSO algorithm. As an objective evaluation metric, the PSD quantifies the differences in denoising performance and signal fidelity among different methods, providing a basis for selecting the optimal denoising method.

Figure 8 illustrates that the PSD of the original noisy signal is higher in the low frequency range, indicating the presence of significant low-frequency noise within the signal. As the frequency increases, the power gradually decreases, although some noise remains in the high frequency range. After processing the signal with a wavelet transform, the PSD in the low frequency region is significantly reduced, demonstrating the effectiveness of the wavelet transform in eliminating low-frequency noise. However, the reduction in PSD within the high frequency range is relatively limited.

Subsequently, after the signal is processed using the PSO-enhanced Wavelet Neural Network, the PSD in both low and high frequency regions decreases markedly. Notably, in the low frequency range, this method exhibits excellent performance in removing low-frequency noise. Meanwhile, noise in the high frequency range is also effectively suppressed, resulting in an overall denoising performance that surpasses the wavelet transform alone. By calculating the frequency-domain signal-to-noise ratio (frequency-domain SNR or FD-SNR) between the denoised signal and the original signal, the fidelity of the denoised signal relative to the original in the frequency domain can be evaluated.

For the RTD fluxgate sensor’s induction signals, the FD-SNR can assess the effectiveness of the denoising algorithm in separating signal from noise, particularly in analyzing the suppression of broadband noise and the concentration of energy in specific frequency components. Data analysis results show that compared to using only the wavelet transform, the denoising process optimized by the improved PSO–Wavelet Neural Network enhances the FD-SNR by 7.36%. The calculated FD-SNR values are as follows.

Combined with the above, the method based on the Wavelet Neural Network and the PSO has obvious advantages in denoising. After denoising, the signal is smoother, and the noise burr is less.

### 4.5. Evaluation of Denoising Effect

The induced signals of RTD fluxgate sensors typically exhibit high sensitivity; however, in practical applications, they are susceptible to Barkhausen noise and environmental noise. SNR serves as an effective metric to assess signal quality, reflecting the relative strength between the signal and noise. Hence, SNR and time difference fluctuation are selected to evaluate the denoising capability and accuracy of the algorithm. A higher SNR value and a lower time difference fluctuation indicate a better denoising quality of the induced signal. The formula for calculating SNR is as follows:(16)$${SNR}_{est}=10\cdot {log}_{10}\left(\frac{T_{noisy}}{T_{denoised}}\right)$$
where $T_{noisy}$ represents the total variation of the noise signal and $T_{denoised}$ represents the total variation of the denoised signal.

The denoising algorithms, including the wavelet transform, Wavelet Neural Network, and the combined method of PSO with the Wavelet Neural Network, are applied to denoise the induced signals. The time difference fluctuation and SNR for each method are calculated accordingly. As shown in Table 5, the denoising algorithm combining the PSO with Wavelet Neural Network achieves a significantly higher SNR compared to the other two methods. This approach effectively improves the time difference SNR while reducing the time difference fluctuation caused by noise, demonstrating superior denoising performance.

## 5. Conclusions

This paper provides a detailed description of the RTD fluxgate sensor’s structure and working principles, emphasizing its vulnerability to Barkhausen noise and environmental noise in complex operating conditions. These noise sources severely impact the sensor’s ability to capture accurate geomagnetic data, reducing both measurement precision and operational stability. To address this, we proposed a denoising method for RTD fluxgate induction signals, utilizing a Wavelet Neural Network enhanced by a PSO algorithm. The method incorporates a dynamic adjustment strategy and a compression factor to enhance denoising effectiveness, measurement accuracy, and the algorithm’s convergence speed and stability. Experimental results demonstrate that, compared to traditional wavelet transform techniques, the proposed method improves the SNR by 0.46% and reduces time difference fluctuations by 23.26%, effectively mitigating the impact of noise on time difference detection. Additionally, the proposed method offers faster computational speed and enhanced adaptability, allowing it to adjust dynamically to varying noise environments. This adaptability is crucial for maintaining the RTD fluxgate sensor’s measurement accuracy and stability across diverse settings, ultimately improving the reliability of UAV-based geomagnetic surveys and environmental monitoring in complex terrains.

## Figures and Tables

**Figure 1 sensors-25-00482-f001:**
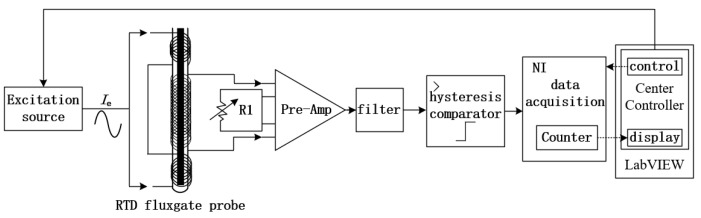
Structure diagram of RTD fluxgate sensitive unit.

**Figure 2 sensors-25-00482-f002:**
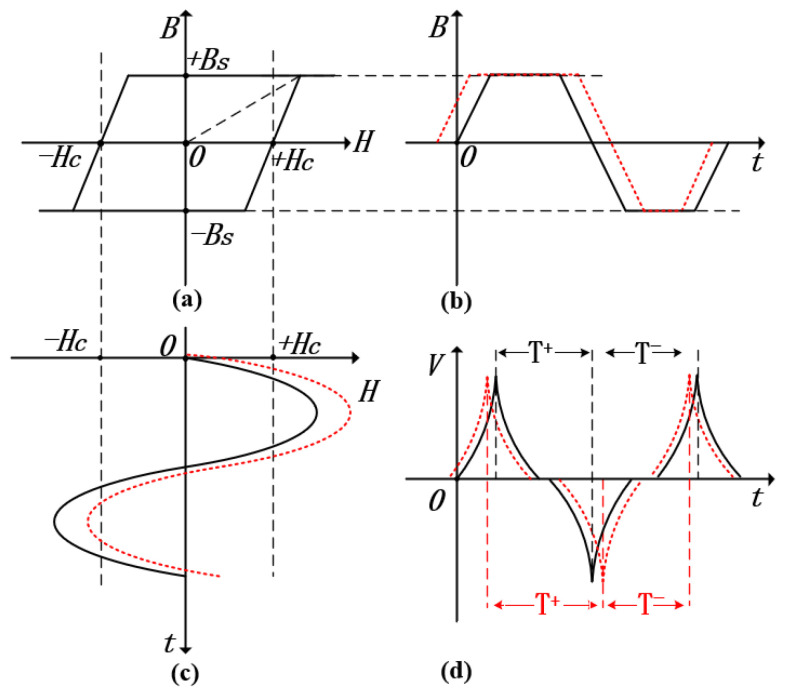
Working principle of RTD fluxgate. (**a**) A hysteresis loop approaching the ideal state; (**b**) Magnetic induction intensity generated in the induction coil; (**c**) exciting magnetic field; (**d**) The induced voltage output.

**Figure 3 sensors-25-00482-f003:**
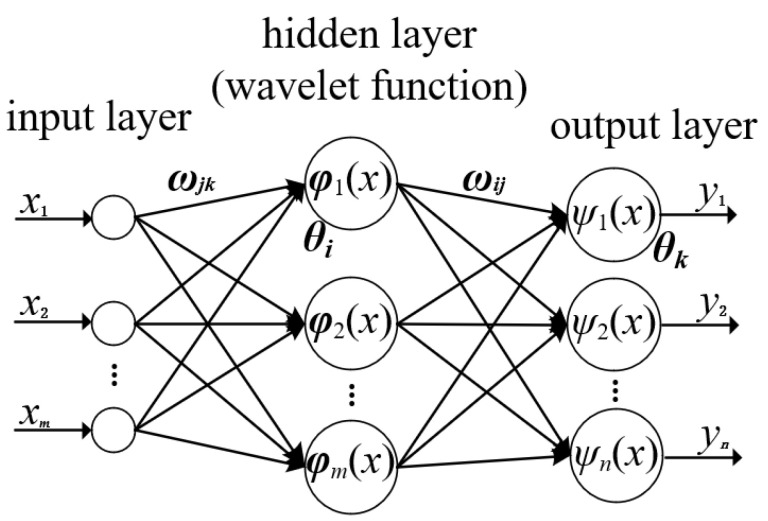
Topological structure of Wavelet Neural Network.

**Figure 4 sensors-25-00482-f004:**
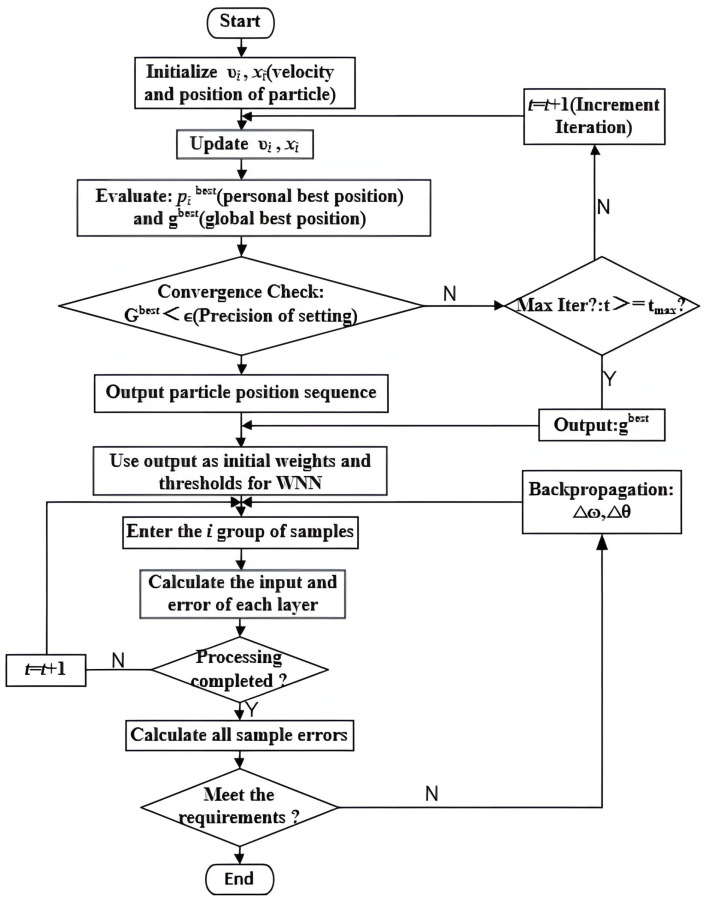
Flow chart of PSO algorithm.

**Figure 5 sensors-25-00482-f005:**
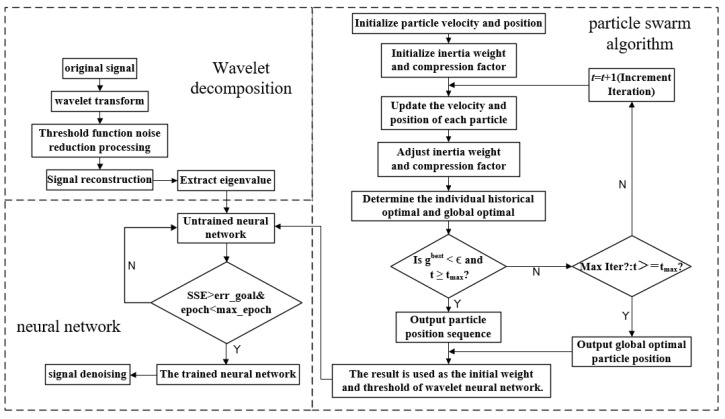
Improved PSO–Wavelet Neural Network flow chart.

**Figure 6 sensors-25-00482-f006:**
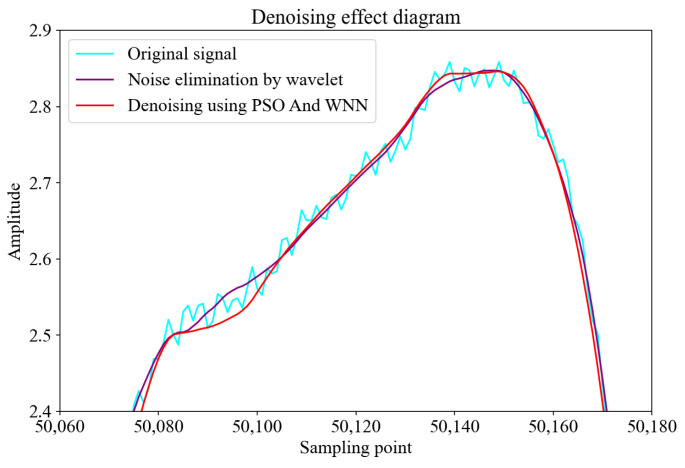
Denoising performance at the signal peak.

**Figure 7 sensors-25-00482-f007:**
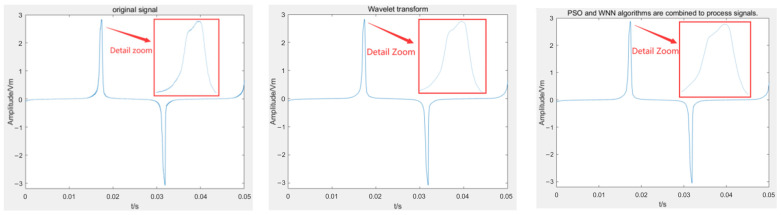
Overall denoising effect.

**Figure 8 sensors-25-00482-f008:**
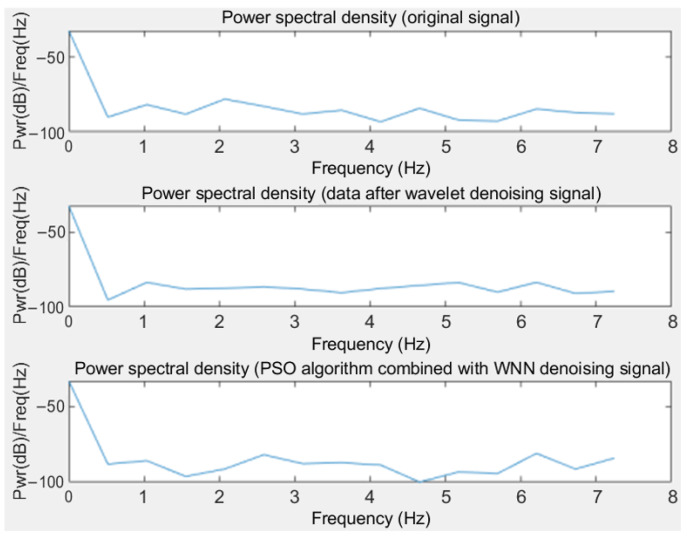
PSD of time difference signals.

**Table 1 sensors-25-00482-t001:** Comparison table of wavelet basis characteristics.

Property	Harr	Mexican Hat	Daubechies
Orthogonality	Yes	No	Yes
Compact Support	Yes	No	Yes
Discrete Wavelet Transform	Possible	Not Possible	Possible
Support Length	1	Finite	2N-1
Symmetry	Symmetric	Symmetric	Near-Symmetric
Vanishing Moments	1	--	N

**Table 2 sensors-25-00482-t002:** db wavelet series SNR and FDV comparison table.

	SNR	FDV
db1	35.86	0
db2	39.31	8.62
db3	39.40	8.35
db4	39.37	8.28
db5	39.29	8.25
db6	39.21	8.20
db7	39.16	8.19
db8	39.10	8.17

**Table 3 sensors-25-00482-t003:** Parameter setting table of Wavelet Neural Network.

Name	Parameter
Number of neurons in the input layer	64
Number of neurons in the hidden layer	64
Number of neurons in the output layer	1
Scale parameter of the connections between the input layer and hidden layer	1.23
Shift parameter of the connections between the input layer and hidden layer	5.8 × 10^−2^
Scale parameter of the connections between the hidden layer and output layer	1.68
Shift parameter of the connections between the hidden layer and output layer	1.4 × 10^−2^
Target accuracy	1 × 10^−4^
Learning rate	1 × 10^−3^
Transfer function	db wavelet basis function
Maximum number of iterations	200

**Table 4 sensors-25-00482-t004:** Sample variance and FD-SNR.

Signal Type	Sample Variance (×10^−3^)	FD SNR (×10^−3^ dB)
Original signal	298	--
Signals after wavelet transform	298	0.83
Signal processed by Wavelet Neural Network	298	−9.01
Improved PSO Wavelet Neural Network to process signals	297	11.27

**Table 5 sensors-25-00482-t005:** Denoising effect evaluation.

Signal Type	Time Difference Volatility (×10^−3^ s)	SNR (dB)
Original signal	215	--
Signals after wavelet transform	200	4.75
Signal processed by Wavelet Neural Network	200	4.74
Improved PSO Wavelet Neural Network to process signals	165	4.77

## Data Availability

Data are contained within the article.

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
