# Peer review of "Research on RTD Fluxgate Induction Signal Denoising Method Based on Particle Swarm Optimization Wavelet Neural Network"

_sensors, 2025, doi:10.3390/s25020482_

Round 1
Reviewer 1 Report
Comments and Suggestions for Authors
Dear authors,
You presented an interesting and important manuscript for studying physical effects using non-destructive methods. The text of the manuscript is well structured and logically presented. The literature review is relevant and fully covers the subject area.
As comments, I would like to add the following.
Please emphasize the novelty of your contribution to the task development.
Please specify the sensors from which material you took into consideration.
It is not clear to me why you use a two-dimensional coordinate system for signal analysis. With your capabilities, you can safely process a 3D signal.
Good luck
Author Response
|
Comments 1: Please emphasize the novelty of your contribution to the task development. |
|
Response 1: Thank you for your valuable suggestion. In response to your comment, we have revised the abstract to better highlight the innovative aspects of our work. Originally, the abstract stated: "This study proposes a denoising method based on a particle swarm optimization (PSO)-enhanced wavelet neural network (WNN), integrating a compression factor and dynamic adjustment strategy to efficiently process RTD-fluxgate signals." We have modified the statement to: "Therefore, we propose a novel denoising method that integrates Particle Swarm Optimization (PSO) with wavelet neural networks, enhanced by a dynamic compression factor and an adaptive adjustment strategy. This approach leverages PSO to fine-tune the wavelet neural network parameters in real time, significantly improving denoising performance and computational efficiency." This revision emphasizes the originality of our approach, particularly the real-time fine-tuning of WNN parameters through PSO, which is a key contribution to improving both denoising performance and computational efficiency.(Modified location: Starting from the sixth line of the abstract in the manuscript, to the tenth line of the abstract) |
|
Comments 2: Please specify the sensors from which material you took into consideration. |
|
Response 2: We have made substantial revisions in the first paragraph of Section 2, "Principle of RTD-Fluxgate and Noise in Sensitive Units," to address the reasoning behind the selection of cobalt-based amorphous materials for the sensor's magnetic core. The revised content now includes detailed information about the magnetic core materials, including cobalt-based amorphous magnetic cores, iron-based nanocrystalline magnetic cores, and iron-based amorphous magnetic cores. We have also provided a comparison of their properties, such as saturation behavior and induced signal characteristics, to justify the material selection for the sensor's sensitive unit. The core of the residence time-difference fluxgate sensor is made of soft magnetic material, which is required to have low coercivity, fast response to magnetic field changes, and characteristics that are easily magnetized by an external magnetic field and demagnetized. Since the core is the core of the sensitive unit and plays a decisive role in the quality of the sensed signal, the selection of the sensitive unit core material will affect the performance of the sensor. Three types of amorphous ribbons with identical geometric dimensions, namely iron-based amorphous (1K101), iron-based nanocrystalline (1K107), and cobalt-based amorphous (1K202), are used as the cores of the sensitive units. When the measured magnetic field Hx=0, the excitation current is 327mA, and the excitation frequency is 200Hz. Based on the measured results from the figure above, it can be observed that the iron-based nanocrystalline (1K107) and cobalt-based amorphous (1K202) magnetic cores reach saturation during the excitation period, with the induced signals exhibiting distinct sharp pulse peaks. However, when using iron-based amorphous (1K101) as the magnetic core, the pulse width of the induced signal fills the entire excitation period, and even when the dynamic magnetic permeability of the core is at its maximum, there is still no distinct sharp pulse in the induced signal. Therefore, it is not suitable as the sensitive element core for a time-difference type fluxgate sensor. Comparing figures 1(b) and 1(c), it is evident that when using a cobalt-based amorphous magnetic core, the sensitive unit outputs the steepest rising edge, the narrowest pulse width, and the highest amplitude of the induced signal, which meets the material selection requirements for the sensitive element core of a time-difference type fluxgate sensor. Where the magnetic core is a cobalt-based amorphous ribbon with thickness, width, and length of 0.025mm, 0.8mm, and 100mm, respectively. The core is placed inside a non-magnetic skeleton with a diameter of 4mm. The length of the excitation coil winding at both ends is 10mm, and the length of the induction coil winding is 40mm. Both the excitation and induction coils use enameled wire with a diameter of 0.062mm, with the number of turns being 100 and 1000, respectively. The skeleton is fixed onto a 110mm by 15mm printed circuit board at both ends using non-magnetic rectangular blocks with dimensions of 20mm by 8mm by 8mm, serving as the sensitive unit of the sensor. We believe these revisions address your concern and improve the clarity of the manuscript. Thank you again for your constructive feedback. |
|
Comments 3: It is not clear to me why you use a two-dimensional coordinate system for signal analysis. With your capabilities, you can safely process a 3D signal. |
|
Response 3: Thank you for your valuable feedback regarding the use of a two-dimensional coordinate system for signal analysis in our manuscript. We understand your suggestion of utilizing a three-dimensional signal processing approach, given the capabilities of the system. However, after careful consideration, we have decided to maintain the use of a two-dimensional coordinate system in this study. The primary reason for this choice is that, in the context of the RTD-fluxgate sensor signals we are analyzing, the most significant variations and noise typically manifest along two dimensions (time and amplitude), which adequately capture the signal characteristics for the denoising approach proposed. A three-dimensional approach, while potentially beneficial in certain cases, would significantly increase the complexity of the processing and may not provide substantial improvements in signal clarity or noise suppression for the specific application of UAV-based geological surveying and environmental monitoring that we are addressing. We hope this explanation clarifies our rationale for the choice of a two-dimensional system. Thank you again for your thoughtful suggestion, and we look forward to your further feedback. |

Reviewer 2 Report
Comments and Suggestions for Authors
Dear Authors,
Thank you for the interesting manuscript. I recommend to take into account some issues:
1. Please introduce the abbreviations in the main text too, not only in the Abstract.
2. The Introduction sections is well-written and the Authors, as I can understand, are the specialists in the field. Nevertheless, I would recommend to add some additional references to the papers of Dr. Andò scientific group (the Authors refer only to [4]), since they are of the pioneers in the field.
3. The magnetic core can be made of amorphous soft magnetic materials without cobalt. See, for example, https://www.sciencedirect.com/science/article/pii/S0924424709000879
4. Please check the figure captions carefully. Figure 1 captions seems to be wrong ("Figure 1. This is a figure. Schemes follow the same formatting").
5. In my opinion, this statement "...the permeability of the core drops to zero" needs to be clarified in more detail, since B = u*uo*H, the static (not dynamic!) permeability of the core decreases after the material magnetic saturation, but cannot reach zero.
6. Please check the manuscript for such misprints: Section 3, "This section may be divided by subheadings. It should provide a concise and precise description of the experimental results, their interpretation, as well as the experimental conclusions that can be drawn".
7. I recommend not to name the sections by any abbreviations ("WNN").
8. I cannot see any profit of using the third level of the subsections. In my opinion, they can be omitted to simplify the text.
9. The manuscript does not fully follow the MDPI style and has to be thoroughly revised for style flaws and typos.
10. Figures resolution seems to be poor and text cannot be easily read. Please improve the figures quality.
11. Please check the higher and lower indices (superscript and subscript), e.g. in the Table 5 ("×10-3s").
12. Since the Manuscript is quite large and complicative, it would be useful for the readers if the Authors add a comparative table to the Results in order to stress the novel achievements comparing to the literature data and the Authors own previously obtained results.
Author Response
|
Comments 1: Please introduce the abbreviations in the main text too, not only in the Abstract. Response 1: We are grateful for the suggestion.In response to the reviewer’s comment regarding the introduction of abbreviations, we have revised the manuscript accordingly. Specifically, we have introduced the abbreviations not only in the Abstract but also in the main text, at the first occurrence of each abbreviation. We believe these changes improve the clarity and readability of the manuscript.(The following abbreviations are defined at their first occurrence in the main text: In the first paragraph of the Introduction: unmanned aerial vehicles (UAVs), residence time-difference (RTD) In the third paragraph of the Introduction: Particle Swarm Optimization (PSO) In Section 4.1: signal-to-noise ratio (SNR)) |
|
Comments 2: The Introduction sections is well-written and the Authors, as I can understand, are the specialists in the field. Nevertheless, I would recommend to add some additional references to the papers of Dr. Andò scientific group (the Authors refer only to [4]), since they are of the pioneers in the field. |
|
Response 2: Thank you for your valuable suggestions regarding this part of the content. 1.Introduction Section: We have added a reference to Dr. Andò's paper [5] in addition to the previously cited work [4], as you suggested, to provide a broader perspective on the pioneering contributions of his research group in this field.(References [4, 5], in the first paragraph of the Introduction.) 2.Sensor Principles Section: We have updated the reference at position [6] to include a more relevant work from Dr. Andò, as the previous reference also cited Dr. Andò’s research. We believe this revision further strengthens the manuscript by highlighting his contributions more explicitly.(References [6], in the second paragraph of the Introduction.) 3.In the same section, the reference at position [17] has been modified for a similar reason. The previous citation was related to Dr. Andò's work, and we have substituted it with a more specific and relevant publication from his group that better aligns with the topic discussed.(The reference [17] in the first paragraph of Chapter 2.) 4.Furthermore, we have added a new reference [18] following the citation at [17]. This addition stems from Dr. Andò's research, which we reviewed while addressing a related issue in the manuscript regarding the use of amorphous soft magnetic materials without cobalt. His work provided valuable insights into this aspect, and we have incorporated it to correct and improve the discussion in the manuscript.(The reference [17] in the first paragraph of Chapter 2.) We believe these updates improve the manuscript and better reflect the important contributions of Dr. Andò’s research group in this area. We hope these changes meet your expectations. |
|
Comments 3: The magnetic core can be made of amorphous soft magnetic materials without cobalt. See, for example, https://www.sciencedirect.com/science/article/pii/S0924424709000879. |
|
Response 3: Thank you for your valuable suggestion. Based on your recommendation, we have made substantial revisions in the first paragraph of Section 2, "Principle of RTD-Fluxgate and Noise in Sensitive Units," to address the reasoning behind the selection of cobalt-based amorphous materials for the sensor's magnetic core. In this revised section, we have provided a more detailed explanation of the benefits and challenges of using cobalt-based amorphous soft magnetic materials, including their superior magnetic properties, which are typically sought for high-sensitivity applications.The reasons are as follows: The core of the residence time-difference fluxgate sensor is made of soft magnetic material, which is required to have low coercivity, fast response to magnetic field changes, and characteristics that are easily magnetized by an external magnetic field and demagnetized. Since the core is the core of the sensitive unit and plays a decisive role in the quality of the sensed signal, the selection of the sensitive unit core material will affect the performance of the sensor. Three types of amorphous ribbons with identical geometric dimensions, namely iron-based amorphous (1K101), iron-based nanocrystalline (1K107), and cobalt-based amorphous (1K202), are used as the cores of the sensitive units. When the measured magnetic field Hx=0, the excitation current is 327mA, and the excitation frequency is 200Hz. Based on the measured results from the figure above, it can be observed that the iron-based nanocrystalline (1K107) and cobalt-based amorphous (1K202) magnetic cores reach saturation during the excitation period, with the induced signals exhibiting distinct sharp pulse peaks. However, when using iron-based amorphous (1K101) as the magnetic core, the pulse width of the induced signal fills the entire excitation period, and even when the dynamic magnetic permeability of the core is at its maximum, there is still no distinct sharp pulse in the induced signal. Therefore, it is not suitable as the sensitive element core for a time-difference type fluxgate sensor. It is evident that when using a cobalt-based amorphous magnetic core, the sensitive unit outputs the steepest rising edge, the narrowest pulse width, and the highest amplitude of the induced signal, which meets the material selection requirements for the sensitive element core of a time-difference type fluxgate sensor. Where the magnetic core is a cobalt-based amorphous ribbon with thickness, width, and length of 0.025mm, 0.8mm, and 100mm, respectively. The core is placed inside a non-magnetic skeleton with a diameter of 4mm. The length of the excitation coil winding at both ends is 10mm, and the length of the induction coil winding is 40mm. Both the excitation and induction coils use enameled wire with a diameter of 0.062mm, with the number of turns being 100 and 1000, respectively. The skeleton is fixed onto a 110mm by 15mm printed circuit board at both ends using non-magnetic rectangular blocks with dimensions of 20mm by 8mm by 8mm, serving as the sensitive unit of the sensor. |
|
Comments 4:Please check the figure captions carefully. Figure 1 captions seems to be wrong ("Figure 1. This is a figure. Schemes follow the same formatting"). |
|
Response 4: We appreciate for the suggestion. We have carefully reviewed all the figure captions in the manuscript, including Figure 1, and have corrected the error. The caption for Figure 1 has been updated to accurately describe the content of the figure, replacing the previous incorrect text with a more appropriate description.(Change "Figure 1. This is a figure. Schemes follow the same formatting." to "Figure 1. Structure diagram of RTD-fluxgate sensitive unit.") |
|
Comments 5:In my opinion, this statement "...the permeability of the core drops to zero" needs to be clarified in more detail, since B = u*uo*H, the static (not dynamic!) permeability of the core decreases after the material magnetic saturation, but cannot reach zero. |
|
Response 5: We are grateful for the suggestion. As you pointed out, the static permeability of the material decreases after magnetic saturation but does not reach zero. In response, we have updated the manuscript to clarify this point. Specifically, we have elaborated on the behavior of magnetic permeability near saturation and incorporated a more comprehensive discussion of the hysteresis loop and its influence on the core's magnetic properties. This revision aims to correct the earlier imprecise wording and ensure that the behavior of the material is accurately described in terms of its static and dynamic permeability.(The second paragraph of Chapter 2 has been revised.) |
|
Comments 6:Please check the manuscript for such misprints: Section 3, "This section may be divided by subheadings. It should provide a concise and precise description of the experimental results, their interpretation, as well as the experimental conclusions that can be drawn". |
|
Response 6: We have carefully reviewed the section for any misprints and made the necessary corrections(The first paragraph of Chapter 3 has been removed:This section may be divided by subheadings. It should provide a concise and precise description of the experimental results, their interpretation, as well as the experimental conclusions that can be drawn.) |
|
Comments 7:I recommend not to name the sections by any abbreviations ("WNN"). |
|
Response 7: Your guidance has been very inspiring to us. In response to your recommendation, we have revised the manuscript by replacing all instances of the abbreviation "WNN" with its full name to ensure clarity and avoid any confusion.(Change all instances of "WNN" in the text to "wavelet neural network) |
|
Comments 8:I cannot see any profit of using the third level of the subsections. In my opinion, they can be omitted to simplify the text. |
|
Response 8: Based on your feedback, we have removed the third-level headings in Chapters 3 and 4 of the manuscript to simplify the structure of the text. |
|
Comments 9:The manuscript does not fully follow the MDPI style and has to be thoroughly revised for style flaws and typos. |
|
Response 9: Thank you for pointing out the issues with the MDPI formatting in the manuscript. We have thoroughly revised the manuscript to ensure it complies with the formatting requirements of MDPI.The specific modifications are as follows: 1.The font and paragraph formatting for all figure captions (As shown in Figure 1:Figure 1. Structure diagram of RTD-fluxgate sensitive unit.) have been adjusted as per the guidelines. 2.The font and paragraph formatting for all table captions (As shown in Table 1:Table 1. Comparison table of wavelet basis characteristics.) have also been corrected, and we have carefully checked the manuscript to ensure no formatting issues have been overlooked or missed. |
|
Comments 10:Figures resolution seems to be poor and text cannot be easily read. Please improve the figures quality. |
|
Response 10: Thank you for pointing out the issues with the number formatting in the manuscript.In response to your comment, we have made the following changes: 1.We have ensured that all numbers with distinguishable values retain two decimal places. 2.For cases where two decimal places were insufficient to distinguish between values, we have opted to retain three decimal places, using scientific notation (e.g., ×10-3) for clarity. We hope these adjustments improve the overall quality and readability of the figures. (The main revisions involve the numbers in the abstract, conclusion, and Tables 2 to 5) |
|
Comments 11:Please check the higher and lower indices (superscript and subscript), e.g. in the Table 5 ("×10-3s"). |
|
Response 11: We have carefully reviewed the manuscript and corrected any formatting issues related to the higher and lower indices to ensure consistency and accuracy throughout the document.(The main modification was to the format of the superscripts in Tables 3 to 5) |
|
Comments 12:Since the Manuscript is quite large and complicative, it would be useful for the readers if the Authors add a comparative table to the Results in order to stress the novel achievements comparing to the literature data and the Authors own previously obtained results. |
|
Response 12: Thank you for your helpful suggestion to include a comparative table in the Results section to highlight the novel achievements of our work compared to the literature and our previously obtained results. We understand the value of such a table in enhancing the clarity and impact of the manuscript. However, after careful consideration, we have found that it is challenging to provide a comprehensive comparative table that adequately captures all the novel aspects of our work in relation to the literature and our prior results. The complexity and diversity of the data presented in the manuscript, combined with the need for careful alignment between various experimental conditions, make it difficult to condense the information into a table format without oversimplifying the findings. Instead, we have provided a more detailed discussion of our results in the relevant sections of the manuscript, emphasizing the key innovations and how they compare with existing literature. We believe this approach preserves the necessary depth and clarity of the analysis. We hope this explanation is acceptable, and we are open to any further suggestions you may have. |

Round 2
Reviewer 2 Report
Comments and Suggestions for Authors
Dear Authors,
Thank you for addressing all my comments.
In my opinion, the Manuscript is now suitable for publication.